# Dynamic Changes of Microbiome with the Utilization of Volatile Fatty Acids as Electron Donors for Denitrification



Okkyoung Choi [1,†], Se-jin Cha [1,†], Hyunjin Kim [1], Hyunook Kim [2] and Byoung-In Sang [1,*]

1   Department of Chemical Engineering, Hanyang University, 222 Wangsimni-ro, Seoul 04763, Korea; okgii77@hanyang.ac.kr (O.C.); ckck2@naver.com (S.-j.C.); hyunjinkim@hanyang.ac.kr (H.K.)
2   Department of Environmental Engineering, University of Seoul, 163 Seoulsiripdae-ro, Seoul 02504, Korea; h_kim@uos.ac.kr
*   Correspondence: biosang@hanyang.ac.kr; Tel.: +82-2-2220-2328
†   The first two authors listed share first authorship.

**Abstract:** Volatile fatty acids can be used as carbon sources for denitrification and are easily supplied as by-products from the anaerobic digestion of waste materials. Nitrification and denitrification processes were carried out in a single reactor feeding volatile fatty acids as electron donors and the changes in microbial communities in the reactor were investigated. The microbial communities in the alternating aerobic and anoxic systems were different, and their structure flexibly changed within one reactor. *Bacteroidetes* and *Firmicutes* were highly distributed during denitrification, whereas *Proteobacteria* was a major phylum during nitrification. In addition, in the denitrification system, the microbial community was substrate dependent. It showed the sequential nitrogen removal in one reactor and the microbial community also followed the change of environmental condition, cyclic nitrification, and denitrification.

**Keywords:** denitrification; fatty acids; high-strength nitrogen wastewater; cyclic nitrification and denitrification; microbial community structure





## 1. Introduction

Conventionally, nitrogen is removed from wastewater through nitrification and denitrification processes. Nitrifying bacteria are aerobic autotrophs that use oxygen as an electron acceptor, whereas denitrifying bacteria are mostly anoxic heterotrophs and use nitrate. Since most wastewater does not have enough reduced-organic compounds, additional carbon sources (electron donors) need to be supplemented for complete denitrification. A variety of electron donors have been tested for autotrophic, or heterotrophic denitrification, e.g., hydrogen [1], thiosulfate [2], sulfate [3], electrode [4], Fe(0) [5], poly-β-hydroxybutyrate [6], alcohol [7–9], and methane [10,11]. Notably, volatile fatty acids are by-products of anaerobic digestion and they are excellent electron donors for denitrification [12–14]. In addition, changes in the structure of a denitrifying microbial community are strongly dependent on the electron donors [15].

To save energy, simultaneous nitrification and denitrification (SND) processes have been used to treat wastewater containing a high amount of nitrogen. The gradient of the oxygen concentration in biofilms allows nitrifying (surface) and heterotrophic denitrifying bacteria (interior) to coexist [16]. Sequential batch reactors (SBRs) have also been used for simultaneous nitrification and denitrification [17]. However, the transition of the microbial community structure between nitrification and denitrification has not been well defined.

In this study, volatile fatty acids were supplied to wastewater with a high nitrogen concentration to investigate the feasibility of their performance as electron donors and to alternate nitrification and denitrification processes for the treatment of wastewater with a high nitrogen concentration. An alternating aerobic and anoxic system was used to initiate nitrification with oxygen present in the head space without additional aeration.

Sequential denitrification was evaluated after oxygen depletion. In addition, the aerobic and anaerobic system interactions of nitrogen removal and microbial community structure were investigated.

## 2. Materials and Methods

### 2.1. Nitrification and Denitrification Experiments

For the nitrification and denitrification experiments, sludge was collected without any treatment from an aeration tank of a local municipal wastewater treatment plant, Jungrang (Seoul, Korea). The sludge was fully aerated to remove residual organics (data not shown). After centrifugation, 20 mL of collected sludge was added to a 250 mL serum bottle as a microbial inoculum for nitrogen removal. Then, 130 mL of medium was added to the bottle; the medium was prepared as follows (per liter of distilled water): $MgSO_4$, 9.5 mg; $CaCl_2 \cdot H_2O$, 1.2 mg; $ZnSO_4 \cdot 7H_2O$ 12.5 mg; $Na_2MoO_4 \cdot 2H_2O$ 1.3 mg; $MnSO_4 \cdot 4H_2O$ 10.1 mg; $CuCl_2 \cdot 2H_2O$ 2.6 mg; $CoCl_2 \cdot 6H_2O$ 0.3 mg; KCl 1 mg; $FeSO_4 \cdot 7H_2O$ 7.3 mg; and EDTA 9.8 mg. The trace element solution for microbial activity was prepared as previously reported [18]. The oxygen present in ambient air as the headspace in the bottles was used for nitrification, without exogenous addition. The temperature was maintained at 25 °C in a shaking incubator with 150 rpm, and the bottles were placed horizontally to increase the surface area of medium faced with ambient air as the headspace in the bottles (Figure 1). Each cycle started after the addition of nitrogen compounds, a sum of $NH_4^+$ and $NO_3^-$ in the form of $NH_4Cl$ and $NaNO_3$, and the cycle was ended if no nitrogen removal occurred. The nitrification–denitrification cycle was repeated 4 times. The time per cycle was approximately 504 h. To mimic wastewater with high concentration of nitrogenous compounds, the nitrogen concentration of $NH_4^+$ and $NO_3^-$ was 1000 mg/L each. A control bottle with no carbon source addition was used to investigate the effects of volatile fatty acid on nitrogen removal via denitrification as electron donors. Acetate and butyrate were used as model compounds of volatile fatty acids and were added independently corresponding to the ratio of COD/N: 1, 2, 3, and 4 for acetate (0.92 g acetate/g COD, Equation (S1) in the Supplementary Materials); 0.5, 1, 2, and 3 for butyrate (0.54 g butyrate/g COD, Equation (S2)). pH was controlled with a phosphate/bicarbonate buffer system to pH 7 ± 0.5. Biomass was analyzed for the volatile suspended solids (VSS) content following the standard methods [19]. No additional carbon sources were used except for acetate or butyrate. Before the addition of acetate and butyrate, the sludge was fully aerated to remove residual organics (data not shown) and the oxygen consumption was only due to nitrification.

### 2.2. Analysis of Nitrogenous Compounds and Chemical Oxygen Demand (COD)

The nitrogen concentrations of ammonium, nitrate, and nitrite were determined reflectometrically using test strips (Merck, Darmstadt, Germany). Each sample was diluted to the appropriate concentration of nitrogen required by the test reagents, and the nitrogen concentration was measured using a RQflex 10 reflectometer (Merck, Darmstadt, Germany). The COD of acetate or butyrate was analyzed using the USEPA-approved dichromate COD method with a commercial reagent kit (Hach Korea, Seoul, Korea) and its absorbance was measured using a Hach spectrometer (DR3900). The supernatant of centrifuged samples was analyzed following the standard methods [19].

### 2.3. Volatile Fatty Acids Analysis

The concentrations of acetate and butyrate were analyzed by GC-FID (Agilent Technologies 7890A Network GC System, Santa Clara, CA, USA) with a HP-Innowax column (30 m × 250 μm × 0.25 μm, Agilent Technologies, Santa Clara, CA, USA). After centrifuging the samples, the pH of the supernatants was reduced below pH 4 using 10% (*v/v*) phosphoric acid before analysis. The temperature of the injector and detector was set at 250 °C. The column temperature was 50 °C initially and was then ramped up to 190 °C at 10 °C/min. Nitrogen was used as the carrier gas at 2 mL/min.

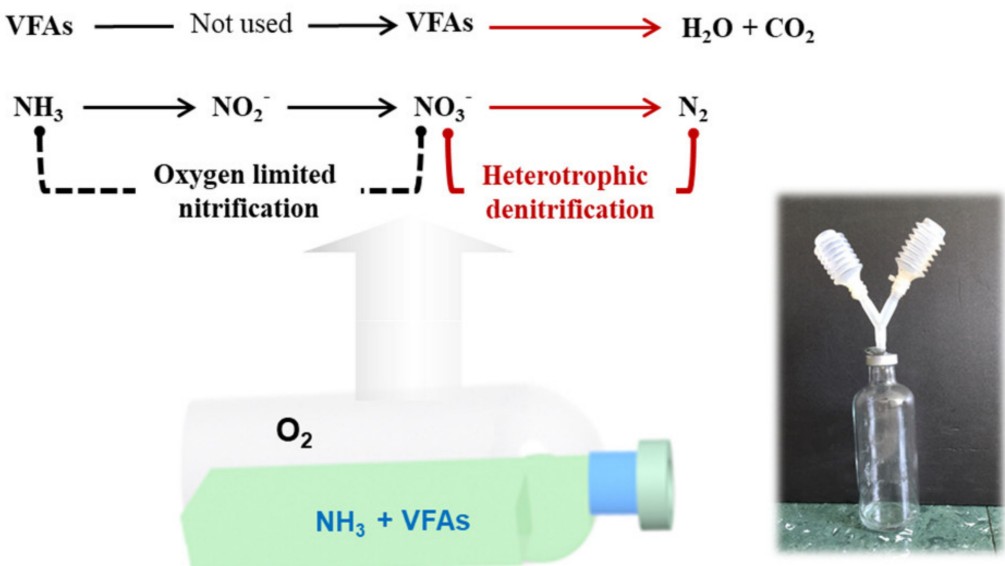

**Figure 1.** Simultaneous nitrification and denitrification system. Flow of nitrogen removal and target analysis in each step, with repeated cycles. Biomass sample was taken from the highest COD/N ratio bottle. The inserted photo is a reactor used in the experiment connected with spring tubes designed in consideration of the reaction that creates negative pressure ($4NH_3 + 8O_2 + 5C \rightarrow 2N_2 + H_2O + 5H^+ + 5HCO_3^-$).

### 2.4. Gas Analysis

The consumption of oxygen for nitrification and production of nitrogen gas through denitrification were measured by a GC-TCD (Agilent technology 7890A Network GC system, Santa Clara, CA, USA). The temperatures of the oven, injector, and detector were 40 °C, 100 °C, and 200 °C, respectively. ShinCarbon ST-micropacked columns (Restek, Bellefonte, PA, USA) were used, and the carrier gas was Ar with a flow rate of 10 mL/min.

### 2.5. Microbial Community Analysis Using Pyrosequencing

The microbial community for nitrification and denitrification were analyzed by pyrosequencing. Samples for community analysis were collected from the highest COD/N ratio samples among the tested bottles containing either acetate or butyrate. Because alternating nitrification/denitrification spontaneously occurred in a single reactor according to the amount of oxygen, unfavorable condition, i.e., no oxygen for nitrifying bacteria or oxygen exposure for denitrifying bacteria, could be a factor in determining the microbial community for nitrogen removal. Therefore, the sample was analyzed at the end of each cycle; nitrification at the end of the 1st cycle (1N), denitrification at the end of the 1st cycle (1D), nitrification at the end of the 2nd cycle (2N), and denitrification at end of the 2nd cycle (2D). In addition, final samples from the skipped nitrification, i.e., no nitrification, were collected to distinguish microbial community structures in denitrification from simultaneous nitrification and denitrification. Electron donors for denitrification were denoted as A for acetate and B for butyrate. The experiment without the addition of a carbon source was used as the control (IF). The microbial community of initial sludge (IS) from the wastewater plant was analyzed for a comparison. Total DNA was extracted with a Power Soil$^{TM}$ DNA isolation kit (Mo Bio Lab., Carlsbad, CA, USA) as described in the manufacturer's instructions. The 16S rRNA genes were amplified (Roche 454 GS FLX Titanium) using bar-coded universal primers for each sample. A universal bacterial primer was used for 16S rRNA gene amplification (27F: AGA GTT TGA TCM TGG CTC AG, 518r: WTT ACC GCG GCT GCT GG). Amplification was carried out under the following conditions: initial denaturation at 95 °C for 5 min, followed by 30 cycles of denaturation at 95 °C for 30 s, primer annealing at 55 °C for 30 s, and extension at 72 °C for 30 s, with a final elongation at 72 °C for 5 min. The amplified products were purified with a QIAquick

PCR purification kit (QIAGEN, Valencia, CA, USA). The obtained readings from different samples were sorted by the unique barcode for each PCR product. The sequences of the barcode, linker, and primers were removed from the original sequencing reads. Any reads containing two or more ambiguous nucleotides, low quality score (average score < 25), or reads shorter than 300bp, were discarded. The diversity was calculated by setting the cutoff value of assigning a sequence to a phylotype at $\geq$97% similarity. Potential chimeric sequences were detected with the Bellerophon method, which compares the BLASTN search results of the forward half sequence and the reverse half. Reads were compared against the EzTaxon-e database (http://eztaxon-e.ezbiocloud.net (accessed on 29 May 2017)) [20]. The overall phylogenetic distance between communities was estimated using the Fast UniFrac [21]. Principal component analysis (PCA) was performed using MATLAB software (R1026b). Pyrosequencing reads generated in this study are available in the EMBL SRA (Sequence Read Archive) database under the accession numbers SUB9562319 under bioproject PRJNA726375.

## 3. Results

### 3.1. Oxygen-Limited Nitrification

The nitrification efficiency did not differ depending on whether acetate and butyrate were added and showed almost the same value in all experiments (Table 1). Approximately 750 mg/L $NH_4^+$-N was removed, and ammonium removals by nitrification stopped around the $O_2$ depletion point in the headspace of the bottle (Figure S1). During nitrification, acetate and butyrate were a little consumed, indicating that nitrifying bacteria mainly consumed oxygen inside the bottle (Figure 2). In the conditions of a very low COD/N ratio and high concentration of ammonia and fatty acids, the activity of heterotrophs was depressed more than that of nitrifiers, so degradation of fatty acids occurred relatively less under nitrification conditions. After the oxygen inside the bottle was exhausted, since there was no additional oxygen supplied, nitrification could not continue, and ammonium could not be completely consumed under all conditions (Table 1). The nitrification efficiency ($NH_4^+$-N removal) was not different even under volatile fatty acid-supplement conditions ($p < 0.05$) and heterotrophic nitrification rarely occurred in all conditions.

**Table 1.** Nitrogen removal in nitrification and denitrification under various COD/N ratios using acetate and butyrate as electron donors for denitrification. The number in parentheses indicates the standard deviation.

| COD/N Ratio | $NH_4^+$-N Removal (%) | $NH_4^+$-N Accumulation (ppm) | $NO_2^-$-N Accumulation (ppm) | $NO_3^-$-N Removal (%) | $NO_3^-$-N Removal Rate (ppm/h) |
|---|---|---|---|---|---|
| 0 | 72.4 ± 0.04 | 270.5 ± 0.5 | 60 ± 0 | 0 | 0 |
| Acetate | | | | | |
| 1 | 73.1 ± 0.47 | 259.5 ± 4.5 | 64.5 ± 0.5 | 27.3 | 6.35 ± 0.05 |
| 2 | 73.5 ± 0.08 | 266 ± 0 | 66 ± 4 | 56.5 | 9.05 ± 0.05 |
| 3 | 74.4 ± 0.70 | 268 ± 6 | 63 ± 1 | 79.9 | 11 ± 0.01 |
| 4 | 74.0 ± 1.29 | 255 ± 11 | 65 ± 0 | 100.0 | 11.45 ± 0.25 |
| Average (s.d.) | 73.8 ± 0.5 | 262.1 ± 5.2 | 64.6 ± 1.1 | | |
| Butyrate | | | | | |
| 0.5 | 73.7 ± 0.56 | 274.5 ± 2.5 | 65.5 ± 1.5 | 30.3 | 5.5 ± 0 |
| 1 | 70.9 ± 0.33 | 286.5 ± 7.5 | 68.5 ± 0.5 | 66.3 | 9.3 ± 0 |
| 2 | 70.7 ± 0.33 | 282 ± 9 | 64 ± 0 | 100 | 11.2 ± 0.8 |
| 3 | 73.3 ± 0.26 | 271 ± 4 | 68.5 ± 0.5 | 100 | 11.5 ± 0 |
| Average (s.d.) | 72.2 ± 1.4 | 278.5 ± 6.1 | 66.6 ± 1.9 | | |

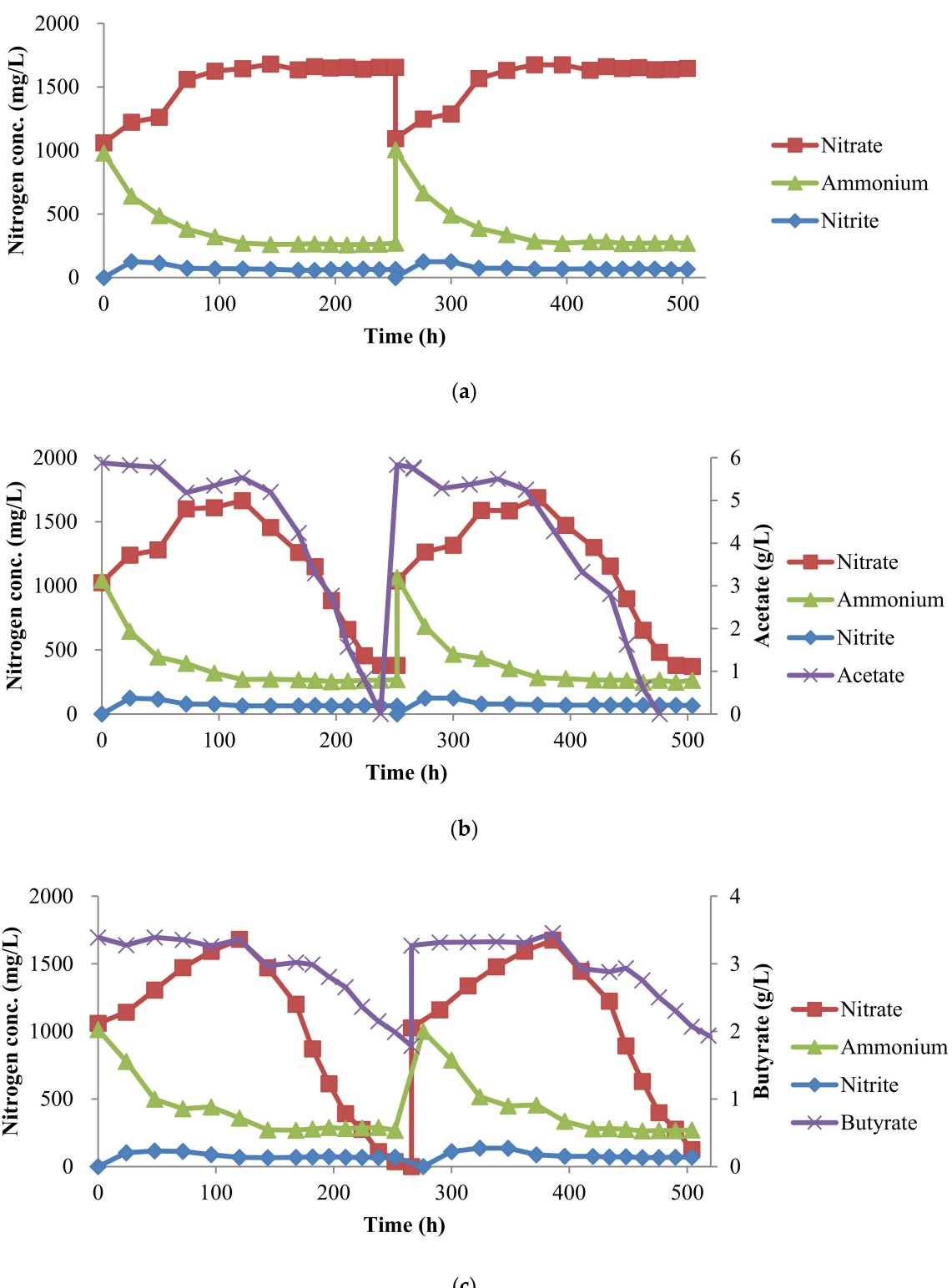

**Figure 2.** Nitrogen removal with (**a**) no exogenous carbon addition; (**b**) acetate addition; and (**c**) butyrate addition at COD/N = 3.

### 3.2. Denitrification with Volatile Fatty Acids

Acetate and butyrate were used as typical volatile fatty acids for the electron donors of heterotrophic denitrification with different amounts according to the ratio of COD/N at fixed nitrogen concentrations, i.e., 1000 mg/L $NH_4^+$-N and 1000 mg/L $NO_3^-$-N. Acetate

and butyrate were not utilized during nitrification and their consumption started with nitrate removal during denitrification (Figure 2b,c). At the same COD/N ratio, butyrate was more efficient for denitrification than acetate (Figure 2b,c and Table 1); at COD/N = 3, the nitrate removal efficiencies by acetate and butyrate were 79.9% and 100%, respectively. Throughout the experiments, VSS concentration was $420 \pm 20$ mg/L, and it did not differ significantly. In denitrification using acetate and butyrate, the nitrogen removal rate was 0.66 g $NO_3^-$-N/g VSS per day, at a 4 COD/N ratio. The values are similar to the previously reported values using acetate and butyrate, 0.13–0.66 [14,22–24], 0.01–0.80 g $NO_3^-$-N/g VSS per day [24–27], respectively. In Figure 3, the equivalent amounts of COD of acetate and butyrate needed for denitrification per unit of nitrate-N were 4.8 and 2.3, respectively; that is, 1 g of nitrate required 1 g of acetate and 0.3 g of butyrate for denitrification. Because volatile fatty acids have different electron equivalents (8 $e^-$/acetate, 20 $e^-$/butyrate, Equations (1) and (2)), different amounts of volatile fatty acids were required as electron donor for denitrification. Due to the complete denitrification with supplied electron donors, acetate and butyrate, accumulated nitrite was not removed during denitrification (Figure 2).

$$5C_2H_3O_2^- + 8NO_3^- + 13H^+ \rightarrow 10CO_2 + 4N_2 + 15H_2O \tag{1}$$

$$5C_4H_7O_2^- + 20NO_3^- + 25H^+ \rightarrow 20CO_2 + 10N_2 + 30H_2O \tag{2}$$

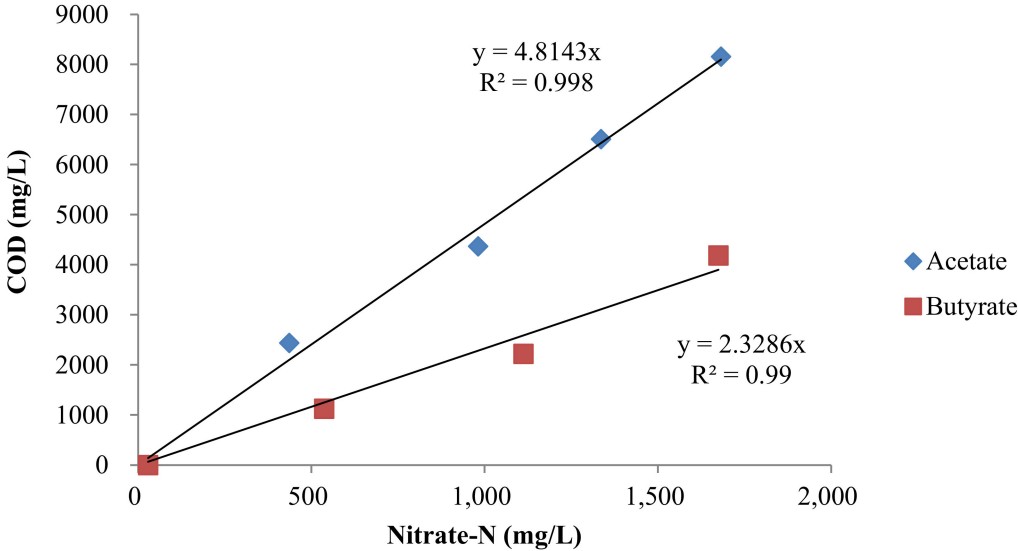

**Figure 3.** COD-based amounts of acetate and butyrate needed for nitrate-N removal.

*3.3. Change of the Microbial Community under Consecutive Aerobic–Anaerobic Conditions for Nitrification and Denitrification*

During consecutive aerobic–anaerobic conditions for nitrification and denitrification, the distribution of nitrification-related bacteria showed that nitrite-oxidizing bacteria (NOB) were more abundant than ammonia-oxidizing bacteria (AOB) (Figure S2). NOB are known as better oxygen competitors than AOB, especially under the limited conditions of oxygen concentration [28]. However, the nitrifying bacteria (i.e., AOB and NOB) disappeared in samples not exposed to aerobic conditions (AD and BD, Figure S2). *Nitrosomonas* and *Nitrospira* were the predominant AOB and NOB, respectively. However, the abundance of the nitrifying bacteria in the microbial community was not high; it was not over 2% (Figure S2).

The community structure of the 1st acetate nitrification (A1N) was relatively similar to that of the no volatile fatty acid condition, whereas community structure of the 2nd acetate nitrification (A2N) was relatively similar to that of butyrate nitrification (B1N, B2N) (Figures 4a and 5a). The 2nd acetate nitrification (A2N) increased the abundance

of *Firmicutes* (4.1%, Figure 4a). Butyrate nitrification further increased the abundance of *Firmicutes* up to 10.5% (B1N) and 12.5% (B2N), respectively. Additionally, *Bacteroidetes* increased in A2N (21.1%), B1N (22.7%), and B2N (23.4%) compared with the no volatile fatty acid control (14.6%).

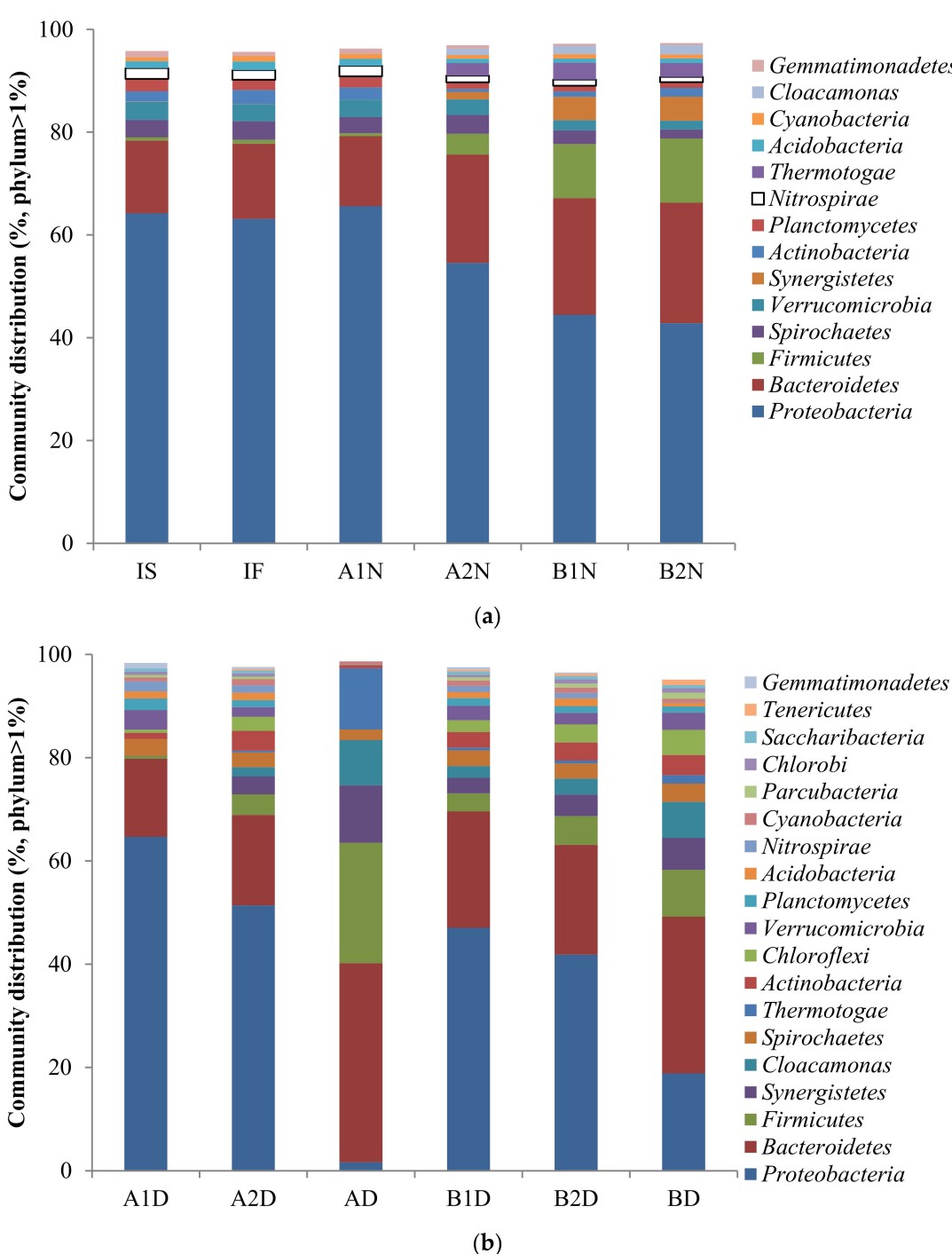

**Figure 4.** The phylum distribution of microbial communities in (**a**) nitrification and (**b**) denitrification processes of each sample. A and B indicate acetate and butyrate, respectively, and N and D represent nitrification and denitrification, respectively. Initial sludge (IS) microbial communities were compared to those in final nitrification samples (IF).

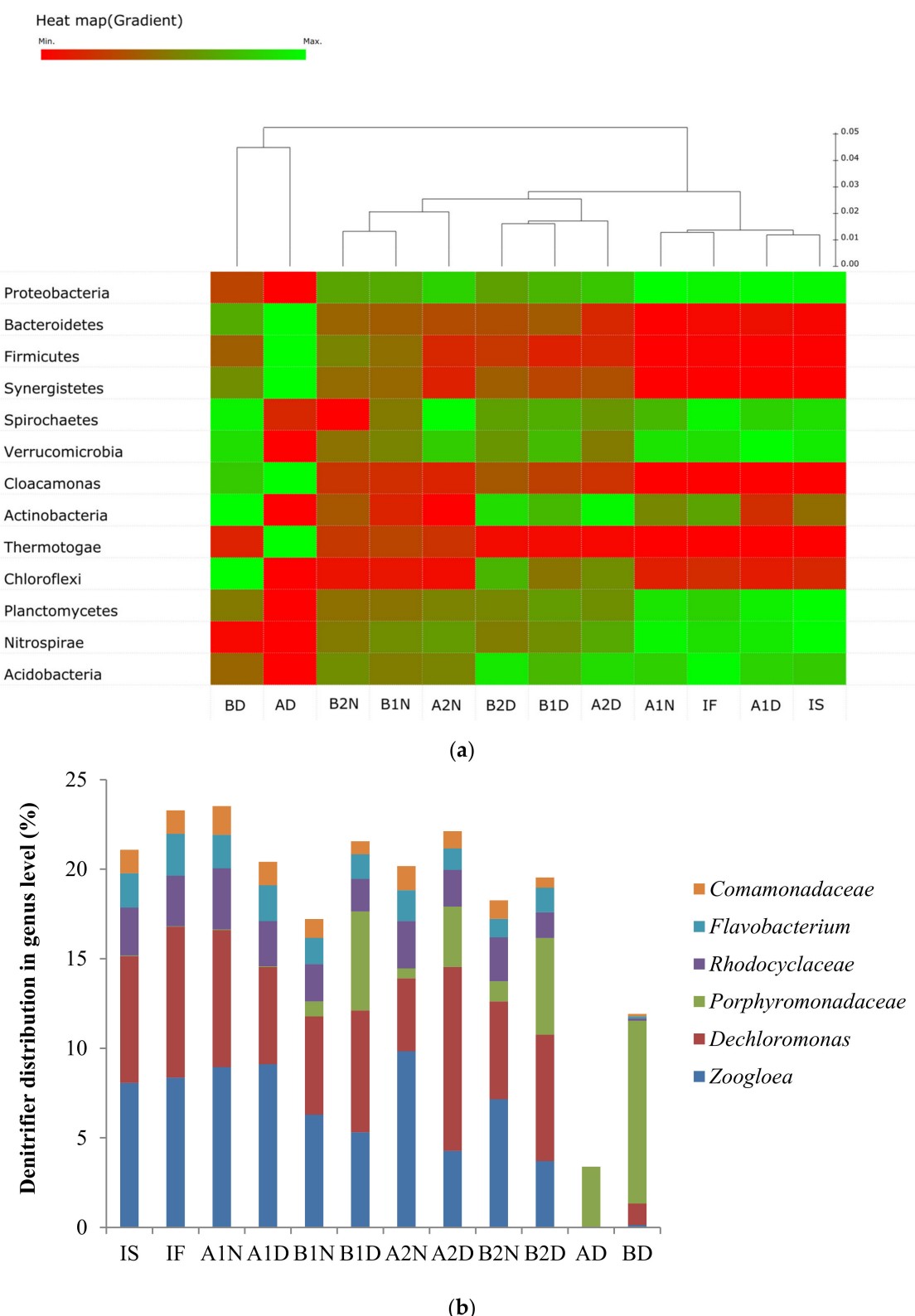

**Figure 5.** The dynamics of microbial communities shown as (**a**) heat maps of the relative abundance of bacteria at the phylum level and (**b**) denitrification-related bacteria (denitrifiers) identified at the genus level. A and B indicate acetate and butyrate, respectively, and N and D represent nitrification and denitrification, respectively. Initial sludge (IS) microbial communities were compared to those in final nitrification samples (IF).

The denitrification community structure that was adapted to the strict anaerobic system was very different from that of the cyclic aerobic–anaerobic system (Figures 4–6). The most abundant phylum shifted from *Proteobacteria* to *Bacteroidetes* (Figure 4b). In addition, the community structure of acetate and butyrate differed; *Bacteroidetes*, *Firmicutes*, *Synergistetes*, *Cloacamonas*, and *Thermotogae* were enhanced in AD, whereas *Acidobacteria* and *Chloroflexi* increased in BD (Figure 4b). *Proteobacteria* was the most abundant phylum in submerged macrophytes [29], a sequencing batch reactor for coking wastewater treatment [30], and a packed-bed bioreactor with a biodegradable polymer [31], whereas Bacteroidetes was the dominant phylum in a swing wastewater anaerobic reactor [32].

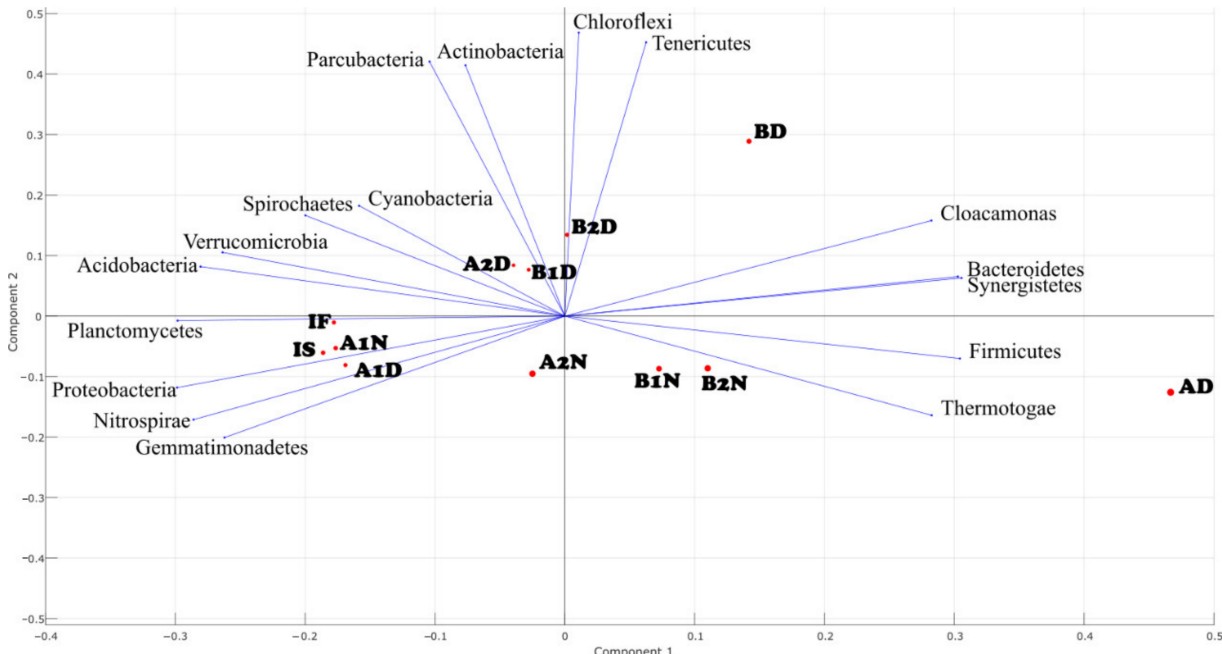

**Figure 6.** PCA plot comparing the phylum-level taxonomic profiles of microbial communities. A and B indicate acetate and butyrate, respectively, and N and D represent nitrification and denitrification, respectively. Initial sludge (IS) microbial communities were compared to those in final nitrification samples (IF).

Interestingly, from the 2nd cycle of acetate, the microbial community structure was relatively similar to that of butyrate-assisted nitrogen removal in both nitrification and denitrification, as shown in the dendrogram (Figure 5a). Acetate and butyrate increased the *Actinobacteria* (mainly *Mycobacterium*) distribution for cyclic denitrification (Figure 5a). *Dechloromonas* was highly distributed among denitrification-related bacteria (denitrifiers), as identified at the genus level (Figure 5b). Denitrifiers represented 7–13.7% of bacteria in the cyclic system, but decreased to 0.7% and 2.1% in AD and BD, respectively, indicating strict anaerobic conditions. At the genus level, *Dechloromonas* was the most abundant in A2D (Figure 5b). A member of *Dechloromonas* was found to be the dominant denitrifying bacteria of the A2N sludge, which showed a relatively high resistance to oxygen [33]. However, *Dechloromonas* decreased (BD) or disappeared (AD) in denitrification only samples, as shown at the genus level (Figure 5b). *Porphyromonadaceae* increased under the fatty acids-amended condition (Figure 5b) and seems to contribute to the denitrification process and carbohydrate utilization in biogas plants [34]. Therefore, fatty acids could be used for other purposes, as well as in denitrification.

## 4. Discussion

The sequential nitrification and denitrification were performed in a single reactor using limited $O_2$ and volatile fatty acids, respectively. The nitrogen removal efficiencies of each cycle were not significantly different from the first cycle (Figure 2). The autotrophic ni-

trifying bacteria were repressed at higher C/N than 10 [35,36]. The cyclic nitrogen removal system can be applied for the treatment of wastewater with high nitrogen concentration such as manure wastewater. Volatile fatty acids such as acetate and butyrate formed a low C/N condition to allow nitrifying bacteria to coexist in the cyclic system. Because most denitrifying bacteria are facultative [37], both nitrifying and denitrifying bacteria could coexist under low C/N conditions. Volatile fatty acids from anaerobic digester [12], food waste leachate [38], and winery wastewater [39] can be applied as electron donors for denitrification and nitrifying bacteria is able to survive by making low C/N conditions in the cyclic system of nitrification–denitrification.

The composition of the microbial community was changed dynamically even under repeated aerobic–anaerobic conditions. Figure 6 shows a principal component analysis (PCA) plot of the phyla with samples. The microbial community structure was dynamic, although the transition between nitrification and denitrification was repeated in one reactor. The distribution of *Thermotogae* seems to increase in butyrate-amended nitrification (B1N and B2N). The butyrate-amended denitrification increased the *Actinobacteria* and *Chloroflexi* distribution. Both have been found in cellulolytic soil [40]. It was reported that clones closely related to deeply branching *Chloroflexi* are involved in butyrate degradation [41]. Volatile fatty acid addition induced an increase of hydrolytic bacteria at the start of anaerobic digestion.

## 5. Conclusions

Both nitrification and denitrification processes were evaluated in a single reactor containing volatile fatty acids; nitrification started with oxygen consumption and denitrification with fatty acid degradation in one reactor. The repeated process did not affect nitrogen removal, and changes in the microbial community were dynamic according to the environment. The cyclic nitrification–denitrification system required 4.8 mg and 2.3 mg of COD acetate and butyrate, respectively, to reduce 1 mg of $NO_3^-$-N to $N_2$. The microbial community structure was substrate dependent. The major phyla of nitrification and denitrification were Proteobacteria and Bacteroidetes, respectively, whereas Actinobacteria increased in cyclic denitrification. This study may increase the understanding of microbial community changes induced by the shift of an electron acceptor ($O_2 \rightarrow NO_3^-$) or electron donor ($CO_2 \rightarrow$ fatty acids) during the nitrogen or carbon removal process. In addition, a single reactor process capable of removing nitrogen through sequential nitrification and denitrification has been proposed.

**Supplementary Materials:** The following are available online at https://www.mdpi.com/article/10.3390/w13111556/s1, Figure S1: The ammonium removal limitation due to oxygen depletion, Figure S2: The distribution of ammonia-oxidizing bacteria (AOB) and nitrite-oxidizing bacteria (NOB) in each sample, Calculations of COD, electron equivalent for acetate and butyrate, Krona charts of each microbiome as HTML files.

**Author Contributions:** Conceptualization, O.C. and B.-I.S.; validation, H.K. (Hyunjin Kim); investigation, S.-j.C.; writing—original draft preparation, O.C.; writing—review and editing, H.K. (Hyunook Kim) and B.-I.S.; supervision, B.-I.S. All authors have read and agreed to the published version of the manuscript.

**Funding:** This work was partially supported by the Korea Institute of Energy Technology Evaluation and Planning (KETEP) grant funded by the Ministry of Trade, Industry & Energy (MOTIE) of the Republic of Korea (No. 20172010000780). This work was supported in part by a grant funded by Hanyang University in the Republic of Korea (HY-201100000000233-N).

**Institutional Review Board Statement:** Not applicable.

**Informed Consent Statement:** Not applicable.

**Data Availability Statement:** Data is contained within the article and Supplementary Materials. Pyrosequencing reads generated in this study are deposited in the EMBL SRA (Sequence Read Archive) database under the accession numbers SUB9562319 under bioproject PRJNA726375.

**Conflicts of Interest:** The authors declare no conflict of interest. The funders had no role in the design of the study; in the collection, analyses, or interpretation of data; in the writing of the manuscript, or in the decision to publish the results.

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
