# Peer review of "Dynamic Changes of Microbiome with the Utilization of Volatile Fatty Acids as Electron Donors for Denitrification"

_water, doi:10.3390/w13111556_

Round 1

Reviewer 1 Report

The objective was to study the application of volatile fatty acids (VFAs) from waste as an external carbon source as a nitrification and denitrification process for the treatment of wastewater containing a high concentration of nitrogen. The study was conducted used a controlled bench-scale system. Food waste and primary sludge were used for anaerobic fermentation to produce VFAs, which were then used as a sole external carbon source with various C/N ratios in the denitrification batch test. 

The use of VFAs produced from anaerobic fermentation as an external carbon source for denitrification has been tested previously by many researchers. The use of carbon resource recovery from wastewater treatment plants is crucial for next-generation wastewater treatment. I support the publication of this work however, the author need to revise and improve the quality of the manuscript.

- The authors did not make clear what are the advantages of using single reactor nitrification and denitrification reactor. The maximum denitrification rate with VFA as an external carbon source 1 mg of NO3-N2 for 4.8 mg acetate and 2.3 mg butyrate, which is one to two orders of magnitudes less than systems that use a two-step process.

- The is no mention of the pH control or buffering of the system. How is the pH monitor as the microbiome species changes?

- There was no manometric tracking as N2 was formed.  

- Discussion section 4 needs a careful edition.

  • Line 231- change to “ different from the first cycle.
  • Line 232 – correct “were was”
  • Line 233 – “ … for the treatment of wastewater…”
  • Line 235- change “Because themost of …”
  • Line 241- the sentence is not clear or not precise. You want to say the composition of the microbiome changes in the course of the process.

Reviewer 2 Report

The present study aims to probe the microbial community dynamics especially for denitrifying populations with feeding acetate and butyrate. Experiments were designed properly, and results were to some extent well documented. However, a few key issues need to be further clarified. Detailed comments are provided hereinafter.   

1) It is somehow illogical that VFAs derived from the AD system were utilized to supply the denitrification given that the VFAs should be converted to methane gas maximumly during the AD process.

2) In Section 3.1, it was stated that acetate and butyrate were hardly consumed, meanwhile, the ammonium oxidation took place actively. This does not comply with the fundamental that the biodegradable COD could more easily access the oxygen in comparison with the ammonium, an inert compound.

3) With the detection of the short region of the 16S rRNA gene, F27/R518, it is unable to get taxon classification at the species level. It is incorrect to state in L187 that Nitrospira defluvii was detected as the representative NOB. The identification of species-level was also mentioned in L214. Please check and revise similar errors through the whole manuscript.

Round 2

Reviewer 2 Report

The 2nd and 3rd comments are not well addressed. 

2) The response is not relevant to the comment.

3) To verify the species, either metagenomic or full-length 16S rRNA gene sequencing is required. The identification method proposed by the authors is to provide/identify the closest species of the representative OTUs rather than consolidate the exact species. 

Authors should address these two issues carefully and properly before receiving a YES for publication. 

Round 3

Reviewer 2 Report

The authors have addressed comments properly. The present version can be accepted.